# Predictive Factors for Decreasing Left Ventricular Ejection Fraction and Progression to the Dilated Phase of Hypertrophic Cardiomyopathy

**DOI:** 10.3390/jcm12155137

**Published:** 2023-08-05

**Authors:** Kakeru Ishihara, Yoshiaki Kubota, Junya Matsuda, Yoichi Imori, Yukichi Tokita, Kuniya Asai, Hitoshi Takano

**Affiliations:** Department of Cardiovascular Medicine, Nippon Medical School, Tokyo 113-0022, Japan; k-ishihara@nms.ac.jp (K.I.); ykubota@nms.ac.jp (Y.K.); jun1984087@nms.ac.jp (J.M.); s9012@nms.ac.jp (Y.I.); yukichi@nms.ac.jp (Y.T.); kasai@nms.ac.jp (K.A.)

**Keywords:** hypertrophic cardiomyopathy, ejection fraction, dilated phase of hypertrophic cardiomyopathy

## Abstract

Patients with hypertrophic cardiomyopathy (HCM) may progress to the dilated phase (DHCM). This study aimed to identify the predictive factors for DHCM progression, including left ventricular (LV) ejection fraction (LVEF < 50%) or decreased LV contraction (LVEF < 60%). The study included 291 patients enrolled in our hospital’s HCM registry who were grouped based on their poststudy LVEF (LVEF of ≥60%, 50–59%, and <50%). Predictive factors of an LVEF of <50% or <60% were determined. Further, the effects of percutaneous transluminal septal myocardial ablation (PTSMA) on long-term systolic LV function and DHCM development were investigated. LVEF was ≥60%, 50–59%, and <50% in 239, 33, and 19 patients, respectively, during the follow-up period (mean: 64.9 months). Multivariate analyses indicated baseline atrial fibrillation (AF), nonsustained ventricular tachycardia (NSVT), and left ventricular diameter at end-systole (LVDs) as significant predictors of DHCM. Using a scoring method based on AF, NSVT, and LVDs, patients with 2 and 3 points had a significantly higher risk of developing DHCM. PTSMA in 78 HCM patients demonstrated no significant effect on long-term LVEF changes or DHCM development. We concluded that AF, NSVT, and LVDs are significant predictors of DHCM development. However, a validation study with a larger population is required.

## 1. Introduction

Hypertrophic cardiomyopathy (HCM) is characterized by left or right ventricular hypertrophy, resulting in impaired left ventricle (LV) diastolic function. HCM, which affects 1 in every 250–500 people, is the most prevalent genetic cardiac disorder [1]. Patients with HCM may progress to the dilated phase (DHCM), in other words, “end-stage HCM”. DHCM is characterized by thinning of a previously hypertrophied ventricular wall, resulting in a decreased LV ejection fraction (LVEF) of <50% and LV dilatation [2]. A recent study estimated that 7.5% of patients with HCM develop DHCM over 15 years [3]. Notably, DHCM is associated with poor prognosis, with approximately 8.4 years (median) to death, requiring an LV assist device or transplantation [3]. Several studies have focused on identifying factors contributing to DHCM development; however, no definitive factors have been identified [4,5,6]. LVEF declines differently in patients with HCM, and no reliable algorithms exist to predict reduced LVEF [5,7]. Neither a method to identify high-risk patients nor a pharmacological intervention to prevent progression to DHCM is currently available. In the future, however, candidate drugs for preventing DHCM may be developed. Identifying high-risk patients, which can make the verification of the drug’s efficacy more efficient, will be helpful, when the phase III clinical trial is conducted.

Most previous studies investigating the factors influencing DHCM did not include patients undergoing septal reduction therapy, such as percutaneous transluminal septal myocardial ablation (PTSMA) [4,5,6]. The nonpharmacological intervention has a direct negative impact on LV contraction. However, the current study included patients who underwent PTSMA because this nonpharmacological treatment is widely performed in patients with significant LV outflow obstruction refractory to adequate pharmacological intervention [8,9]. Thus, treating these cases as unusual is inappropriate. PTSMA was performed in more than one-fourth of the patients with DHCM during the study period. Thus, the impact of PTSMA on long-term LV contractile function, particularly DHCM development, was investigated. Additionally, we concentrated on patients with slightly reduced LVEF, which could be considered a transitional stage to DHCM. Thus, we investigated the factors contributing to a declining LVEF of <50% and 50–59%. The current study aimed to identify the risk factors for DHCM development and for declining LVEF in patients with HCM in the general population, including those with HCM who underwent PTSMA.

## 2. Materials and Methods

### 2.1. Participants

This single-center study included patients enrolled in our hospital’s HCM registry from 1 February 2009 to 31 December 2018. We analyzed the data of 291 patients with HCM after excluding patients who died or dropped out within 2 years (39 patients) (Figure 1). Patients were divided into three groups based on LVEF at the end of the study period: group A (LVEF of ≥60%), group B (LVEF of 50–59%), and group C (LVEF of <50%). We investigated predictors of an LVEF of <50% and 60%. The study protocol adhered to the principles of the Declaration of Helsinki and was approved by the Nippon Medical School Hospital Institutional Ethics Committee (O-2021-006). Informed consent was not required because of the retrospective study design.

### 2.2. Medical History Data

Patient data were gathered from medical records. Echocardiography or cardiac magnetic resonance imaging, which shows a maximal LV wall thickness of ≥15 mm (≥13 mm in patients with a family history of hypertrophic cardiomyopathy), was defined as HCM [10,11]. DHCM was defined as resulting in a decreased LVEF of <50% and LV dilatation [2]. PTSMA was performed for drug-refractory symptomatic hypertrophic obstructive cardiomyopathy (HOCM) with an LV obstruction of ≥50 mmHg, either at rest or after provocation. The procedure of PTSMA is previously described [9]. Hypertension was a systolic blood pressure of ≥140 mmHg, diastolic blood pressure of ≥90 mmHg, and/or current antihypertensive medication intake. Chronic kidney disease was defined as less than an estimated glomerular filtration rate (eGFR) of 60 mL/min/1.73 m^2^ [12,13]. Electrocardiography revealed either paroxysmal or persistent atrial fibrillation (AF). Valvular heart disease was severe tricuspid valve regurgitation, severe mitral valve regurgitation and/or stenosis, and/or severe aortic valve regurgitation and/or stenosis. LVEF was calculated during each echocardiographic study using Teichholz or a modified Simpson’s method. Valvular heart disease severity was assessed using quantitative measurements from transthoracic echocardiography: regurgitation volume, regurgitation jet area, pressure gradient, flow velocity, and effective regurgitant orifice area. All parameters were measured annually and compared with the baseline data. The baseline data comprised pre-electrocardiogram (ECG) and transthoracic echocardiography (TTE). Post-ECG and TTE were performed at the last hospital visit. Two observers assessed all short- and long-axis contrast-enhanced images to determine the dichotomous absence or presence of late gadolinium enhancement by cardiovascular magnetic resonance (CMR). The method of CMR evaluation is previously described [14]. CMR was imaged at the same time as the initial TTE.

### 2.3. Study Protocol

The baseline characteristics of patients in groups A (LVEF of ≥60%), B (LVEF of 60–50%), and C (LVEF of <50%) were compared. We identified and compared the risk factors for DHCM (LVEF of <50%) in group A vs. groups B and C. Additionally, risk factors for developing DHCM or declining LVEF (<60%) were identified. A scoring method based on three parameters detected in the top 3 univariate significant variables was considered. One point was assigned to each risk factor to determine the risk of developing DHCM or declining LVEF (<60%).

### 2.4. Statistical Analysis

Categorical variables are expressed as percentages (%), and continuous variables are expressed as means ± standard deviations. The eligible variables were entered into the multivariable model by stepwise selection to identify potential risk factors for LVEF reduction. Since the number of patients was small in groups B and C, we limited the selection to top 3 variables for multivariate analysis out of the significant variables in the univariate analysis. The cut-off value of each parameter was detected according to the receiver operating characteristic curve. Two-sided *p*-values of <0.05 were considered statistically significant in logistic regression analyses. Statistical Package for the Social Sciences version 21.0 was used for all statistical analyses (SPSS Inc., Chicago, IL, USA).

## 3. Results

LVEF was ≥60% in 239 patients (group A) during the follow-up period (mean: 64.9 months), 50–59% in 33 patients (group B), and <50% in 19 patients (group C). Group C (LVEF of <50%) had a significantly higher male proportion, AF incidence, nonsustained ventricular tachycardia (NSVT), pacemaker implantation (PMI) or implantable cardioverter defibrillator (ICD), serum C-reactive protein (CRP), serum creatinine, eGFR values, and LVDd and LVDs than groups A and B at baseline, as shown in Table 1. Additionally, groups B and C (LVEF of <60%) had significantly higher DM incidences. The rate of declining LVEF was nearly similar in groups B and C (−3.05% ± 2.84% and −3.80% ± 2.43%) and significantly higher than in group A (*p* = 0.011). PTSMA was administered to only one patient in group C and 74 patients in groups A + B (5% vs. 27.2%, *p* = 0.025). Patients with a final LVEF of <60% (groups B + C) had a higher incidence of hospitalization for HF, and patients with a final LVEF of <50% (group C) had significantly higher mortality during the study period. Typical images of TTE and ECG in the three groups are shown in Appendix A.

### 3.1. Factors That Predict LVEF of <50%

Univariate and multivariable analyses were performed to identify the predictive factors for developing DHCM (LVEF of <50%). Male sex, AF, eGFR, NSVT, PMI or ICD, CRP, and LVDs were predictive factors for an LVEF of <50% at the end of the study period in univariate regression analyses (Table 2). AF, NSVT, and LVDs were predictive factors for an LVEF of <50% according to the multivariable analysis (Table 2).

### 3.2. Factors That Predict LVEF of <60%

Univariate and multivariable analyses were performed to identify predictors of a declining LVEF of <60%. Male sex, AF, eGFR, DM, NSVT, PMI or ICD, LVDs, and CRP significantly influenced the declining LVEF in univariate regression analyses (Table 3). The predictive factors in the multivariable analysis were AF, NSVT, and LVDs (Table 3).

### 3.3. Scoring Method

AF, NSVT, and LVDs were significant factors contributing to DHCM development according to the multivariate analysis (Table 2). Hence, we propose a scoring method based on the following three parameters: AF, NSVT, and LVDs of >25.4 mm, with 1 point assigned to each risk factor to determine the risk of developing DHCM or declining LVEF (<60%). Patients with 2 and 3 points had a significantly higher risk of developing DHCM (odds ratio (OR): 5.06; 95% confidence interval (CI): 1.95–13.10; *p* = 0.001 and OR: 48.21; 95% CI: 8.59–270.76; *p* < 0.001, respectively) (Figure 2). The risk of declining LVEF (<60%) was significantly associated with 2 and 3 points (OR: 4.00; 95% CI: 2.09–7.69; *p* < 0.001 and OR: 31.04; 95% CI: 3.65–263.96; *p* = 0.002, respectively) (Figure 3).

Patients with 2 and 3 points had a significantly higher risk of declining LVEF using the scoring method with three parameters. The cut-off value for LVDs, based on the receiver operating characteristic curve, was 25.4 mm. Figure 2 shows the parameter cut-off values.

### 3.4. Effect of PSTMA on LVEF

Patients were divided into three groups based on the study period to investigate the effects of PTSMA on the rate of declining LVEF (ΔLVEF/year): PTSMA before the study period (*n* = 49), PTSMA during the study period (*n* = 30), and no PTSMA (*n* = 212). The rate of LVEF decline/year did not differ between the three groups (Figure 4).

## 4. Discussion

The current study aimed to identify the risk factors for DHCM development (LVEF of <50%) or decreased LV contraction (LVEF of <60%) in patients with HCM. Our results revealed that baseline AF, NSVT, and LVDs are significant predictors of DHCM and declined LV contraction. We concluded that these risk factors were significant predictors of developing DHCM. Further, we incorporated the three risk factors into a simple scoring method to help stratify the risk of developing DHCM.

The onset of the dilated phase is the worst-case scenario in patients with HCM, and the prognosis for patients who progress to DHCM is worse than that for patients with nondilated or even dilated cardiomyopathy [5,15]. Further, the current study demonstrated that patients with DHCM had a significantly worse prognosis. Hence, preventing DHCM is one of the most important issues for cardiologists who treat patients with HCM. Understanding the factors that influence DHCM development and the risk stratification of patients with HCM is crucial for establishing a preventative intervention. A recent study identified [3] several independent risk factors for developing DHCM, including late gadolinium enhancement, sarcomeric variants, and borderline-low baseline LVEF [3]. Another study identified a family history of HCM, younger age at diagnosis, and a thicker wall as risk factors for developing a dilated-hypokinetic phenotype in patients with HCM [4]. The current study demonstrated that LVDs and AF (paroxysmal, persistent, or chronic) before enrollment were significant predictors of DHCM. Many studies [4,16,17,18,19,20] have reported the importance of AF in patients with HCM. AF increases the risk of heart failure, stroke, and sudden death [4,17,20]. AF in patients with HCM is closely associated with significant LV diastolic dysfunction, which disrupts blood inflow to the LV from the LA, resulting in volume and pressure overload in the LA. Hence, AF often coincides with advanced myocardial damage, which causes diastolic dysfunction, followed by subsequent systolic dysfunction.

NSVT was significantly associated with declining LVEF and DHCM development among the predictive factors of sudden cardiac death in patients with HCM identified in recent guidelines ([2,10] ESC guideline). Middle-aged or older patients with HCM have a higher NSVT incidence due to fibrosis and progressive myocyte loss [21], and contractile function declines in parallel with these pathological changes.

LV performance at baseline is one of the predictors of DHCM development [6]. LVDs were the most sensitive echocardiographic parameter, reflecting LV contraction and LV cavity enlargement. Hence, LV contractile function attenuation and LV cavity enlargement result in increased LVDs.

The current study focused on mildly reduced LV contractile function (LVEF = 50–59%) because this status is the transition to DHCM, and a certain percentage of patients with mildly reduced LV contractile function will develop DHCM in the future. However, the declining LVEF rate in those patients was nearly identical to that in patients with DHCM in the current study. AF, NSVT, and LVDs were predictive factors for DHCM. These predictive factors were incorporated in our newly proposed scoring method, which stratifies patients based on their risk for DHCM.

PTSMA is a septal reduction therapy for severe HOCM; however, many patients who are not candidates for surgical myectomy now receive this catheter-based procedure [8,9,22]. The long-term impact of PTSMA on LV contraction, particularly LV development, has not been thoroughly investigated because most previous studies excluded patients receiving septal reduction therapy from DHCM risk analyses [4,5,6]. The present study did not exclude patients with PTSMA because they represented one-fourth of the total patient population. The current study demonstrated that PTSMA does not hasten DHCM development, despite temporarily and mildly reducing LV systolic function immediately after the procedure.

The present study has several limitations. First, this was a single-center retrospective analysis of prospective registry data and included a relatively small sample. Second, the follow-up period may not be long enough to detect the risk of DHCM, and the follow-up time varied among the three groups. Furthermore, we excluded patients who died during the 2 years of follow-up. Third, late gadolinium enhancement on magnetic resonance imaging was not quantified, and we could not determine the significance of DHCM development. Fourth, the predictive factors we examined may be both causes and consequences of LVEF reduction. Thus, the predictive factors mentioned here may actually be more appropriately termed as associated factors. Finally, we propose a new scoring method for stratifying patients based on DHCM risk, but this scoring system needs to be validated in different and larger populations.

## 5. Conclusions

Our data indicate that AF, NSVT, and LVDs at baseline are significant predictors of DHCM development. We propose a new, simple scoring method based on those three factors to stratify the risk of DHCM. However, a validation study including a larger population is required.

## Figures and Tables

**Figure 1 jcm-12-05137-f001:**
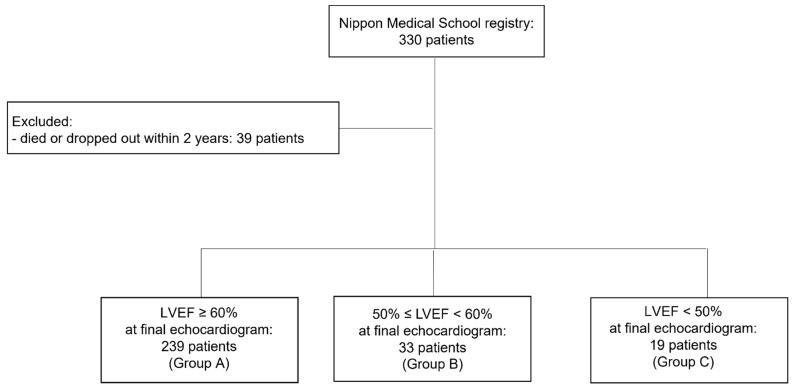
Recruitment procedure. LVEF: left ventricular ejection fraction.

**Figure 2 jcm-12-05137-f002:**
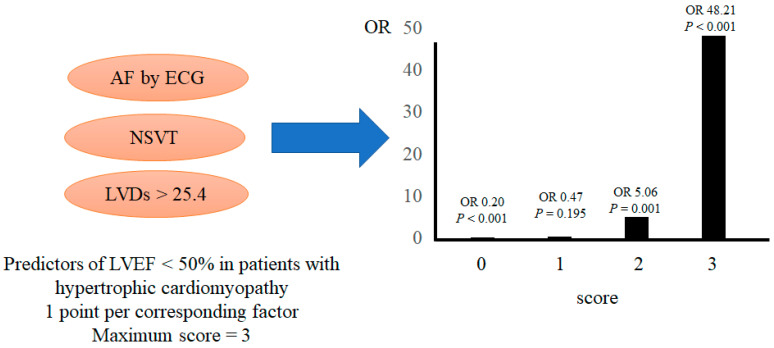
Predictors of an LVEF of <50% in patients with hypertrophic cardiomyopathy. AF: atrial fibrillation; ECG: electrocardiogram; LVDs: left ventricular diameter at the end of systole; NSVT: nonsustained ventricular tachycardia; OR: odds ratio.

**Figure 3 jcm-12-05137-f003:**
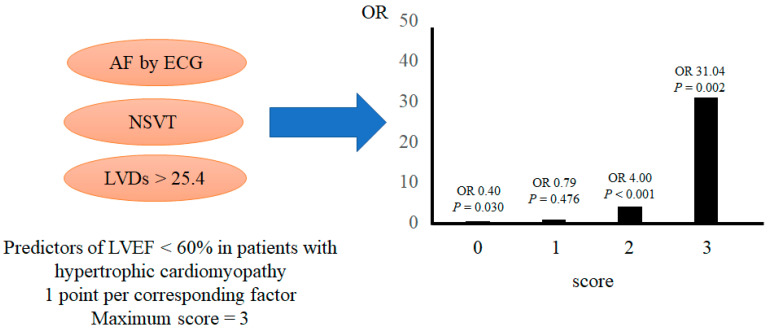
Predictors of an LVEF of <60% in patients with hypertrophic cardiomyopathy. AF: atrial fibrillation; ECG: electrocardiogram; LVDs: left ventricular diameter at the end of systole; NSVT: nonsustained ventricular tachycardia; OR: odds ratio.

**Figure 4 jcm-12-05137-f004:**
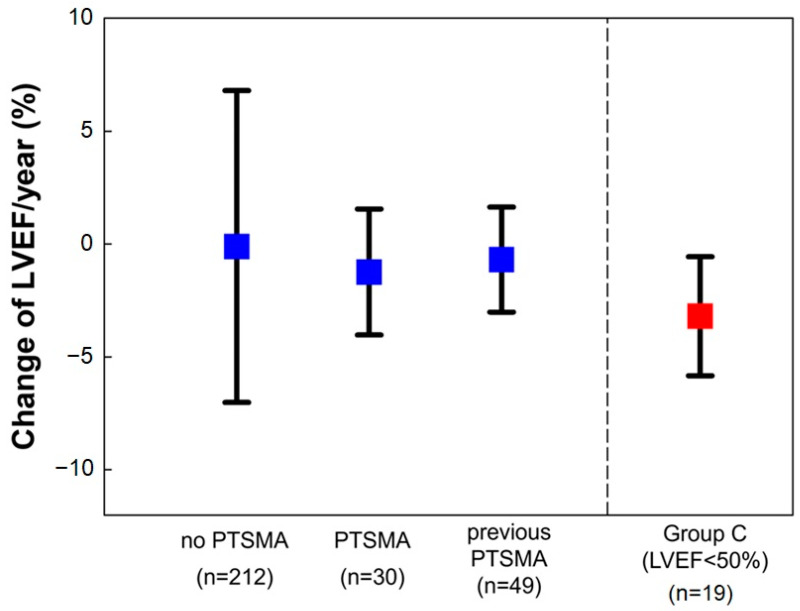
Effects of PTSMA on the rate of LVEF decline. LVEF: left ventricular ejection fraction; PTSMA: percutaneous transluminal septal myocardial ablation. As a reference, the rate of LVEF decline in group C (LVEF of <50% at the end of the study period) is also shown.

**Table 1 jcm-12-05137-t001:** Patient characteristics.

	Group A	Group B	Group C	A + B vs. C	A vs. B + C
Variables	LVEF ≥ 60%	50% ≤ LVEF < 60%	LVEF < 50%	*p*-Value	*p*-Value
	(*n* = 239)	(*n* = 33)	(*n* = 19)		
Age (years)	65.45 ± 14.05	60.58 ± 12.71	64.26 ± 13.66	0.940	0.154
Men, *n* (%)	108 (45)	18 (55)	14 (74)	0.019	0.028
Atrial fibrillation, *n* (%)	26 (11)	6 (18)	11 (58)	<0.001	<0.001
Hypertension, *n* (%)	129 (54)	13 (42)	10 (53)	0.700	0.566
Diabetes mellitus, *n* (%)	31 (13)	13 (42)	6 (33)	0.346	0.010
HOCM, *n* (%)	70 (29)	16 (48)	1 (5)	0.022	0.341
Family history of SCD	13 (5)	1 (3)	2 (11)	0.322	0.925
NSVT, *n* (%)	57 (24)	15 (45)	12 (63)	0.001	<0.001
Laboratory data					
Troponin T, ng/mL	0.024 ± 0.083	0.021 ± 0.019	0.024 ± 0.077	0.897	0.637
Creatinine, mg/dL	0.81 ± 0.26	0.90 ± 0.19	1.22 ± 0.91	<0.001	<0.001
eGFR, mL/min/1.73 m^2^	67.38 ± 17.94	60.81 ± 16.81	57.63 ± 22.53	0.036	0.005
Chronic kidney disease, *n* (%)	81 (34)	18 (55)	9 (47)	0.392	0.024
CRP, mg/dL	0.20 ± 0.53	0.20 ± 0.36	0.59 ± 1.19	0.004	0.127
BNP, pg/mL	291.86 ± 680.35	403.17 ± 418.21	295.10 ± 255.54	0.961	0.484
NT-pro BNP, pg/mL	1006.16 ± 1668.24	1334.80 ± 1467.11	879.50 ± 707.11	0.888	0.766
Pre-ECG					
QRS duration, msec	103.53 ± 18.01	103.94 ± 16.18	109.16 ± 23.89	0.177	0.356
RV5 + V1S, mV	4.26 ± 1.86	3.963 ± 1.875	3.561 ± 1.98	0.173	0.130
CRBBB, *n* (%)	24 (10)	3 (9)	3 (16)	0.433	0.778
Post-ECG					
QRS duration, msec	113.80 ± 28.30	118.48 ± 33.87	113.26 ± 24.41	0.87	0.529
RV5 + V1S, mV	3.33 ± 1.60	2.56 ± 1.36	3.11 ± 1.51	0.749	0.02
CRBBB, *n* (%)	72 (30)	11 (33)	4 (21)	0.377	0.832
Pre-TTE					
LVEF, %	74.48 ± 6.43	70.33 ± 8.00	59.58 ± 7.21	<0.001	<0.001
IVST, mm	14.07 ± 4.84	14.84 ± 4.94	13.95 ± 4.64	0.849	0.561
PWT, mm	10.15 ± 2.12	11.13 ± 3.50	9.32 ± 1.66	0.078	0.430
Maximum thickness	15.27 ± 4.76	14.94 ± 4.64	14.56 ± 4.14	0.820	0.703
LVDd, mm	43.25 ± 6.12	43.97 ± 6.02	47.00 ± 8.49	0.022	0.078
LVDs, mm	24.52 ± 5.06	26.34 ± 5.51	33.00 ± 9.59	<0.001	<0.001
LAD, mm	39.42 ± 7.61	40.94 ± 8.34	43.21 ± 7.99	0.057	0.056
E/A	1.08 ± 1.44	1.01 ± 0.51	1.03 ± 0.54	0.894	0.778
E/e’	14.42 ± 6.14	15.58 ± 10.59	12.5 ± 5.20	0.215	0.937
Severe AS	7 (3)	1 (3)	1 (5)	0.573	0.730
Severe MR	18 (8)	1 (3)	4 (21)	0.028	0.615
Severe TR	15 (6)	1 (3)	2 (11)	0.418	0.891
Post-TTE					
LVEF, %	72.57 ± 6.75	55.18 ± 3.00	39.47 ± 10.88	<0.001	<0.001
ΔLVEF, %	−1.92 ± 8.32	−15.15 ± 7.48	−20.11 ± 12.53	<0.001	<0.001
ΔLVEF/year, %	−0.25 ± 6.37	−3.05 ± 2.84	−3.80 ± 2.43	0.010	0.011
LVDd, mm	42.83 ± 6.08	46.97 ± 5.69	52.68 ± 10.59	<0.001	<0.001
LVDs, mm	25.62 ± 4.94	32.79 ± 4.63	41.37 ± 12.58	<0.001	<0.001
CMR					
LGE, *n* (%)	91 (38)	17 (52)	9 (47)	0.499	0.104
Nonpharmacological therapy					
PTSMA, *n* (%)	62 (26)	16 (48)	1 (5)	0.025	0.347
PMI or ICD, *n* (%)	33 (14)	8 (24)	7 (37)	<0.001	0.008
Ablation for AF, *n* (%)	33 (14)	4 (12)	5 (26)	0.130	0.525
Medication					
β-blocker, *n* (%)	213 (89)	27 (83)	16 (87)	0.891	0.999
ACE-I or ARB, *n* (%)	60 (25)	19 (58)	10 (53)	0.010	0.001
MRA, *n* (%)	5 (2)	13 (42)	7 (40)	<0.001	<0.001
Furosemide, *n* (%)	21 (9)	8 (24)	7 (37)	0.001	<0.001
Azosemide, *n* (%)	12 (5)	2 (8)	1 (7)	0.898	0.722
Torasemide, *n* (%)	7 (3)	2 (8)	2 (13)	0.062	0.048
Events					
Hospitalization due to HF, *n* (%)	24 (10)	6 (20)	6 (32)	0.010	0.060
Death, *n* (%)	12 (5)	1 (3)	6 (32)	<0.01	0.029
Follow-up period, months	56.2 ± 64.8	59.0 ± 33.4	69.6 ± 31.2	0.002	0.011

Data are presented as means ± standard deviations or numbers (percentages). ACE-I = angiotensin-converting enzyme inhibitor; ARB = angiotensin II receptor blocker; AS = aortic valve stenosis; BNP = brain natriuretic peptide; CMR = cardiovascular MRI; CRBBB = complete right bundle branch block; CRP = C-reactive protein; ECG = electrocardiogram; eGFR = estimated glomerular filtration rate; HF = heart failure; HOCM = hypertrophic obstructive cardiomyopathy; ICD = implantable cardioverter defibrillator; IVST = interventricular septum thickness; LAD = left atrial dimension; LGE = late gadolinium enhancement; LVDd = left ventricular end-diastolic dimension; LVDs = left ventricular diameter at end-systole; LVEF = left ventricular ejection fraction; MR = mitral valve regurgitation; MRA = mineralocorticoid receptor antagonist; NT-pro BNP = N-terminal prohormone of brain natriuretic peptide; SCD = sudden cardiac death; NSVT = nonsustained ventricular tachycardia; PMI = pacemaker implantation; PTSMA = percutaneous transluminal septal myocardial ablation; PWT = posterior LV wall thickness; SD = standard deviation; TTE = transthoracic echocardiography; TR = tricuspid valve regurgitation.

**Table 2 jcm-12-05137-t002:** Univariate and multivariable analyses for predictors of LVEF < 50%.

	Univariate Analysis		Multivariable Analysis	
	HR (95% CI)	*p*-Value	HR (95% CI)	*p*-Value
Age	0.99 (0.96–1.03)	0.946		
Male sex	3.29 (1.15–9.39)	0.026		
Atrial fibrillation	10.2 (3.79–27.09)	<0.001	14.00 (4.42–44.38)	<0.001
eGFR †	0.97 (0.95–0.99)	0.041		
Diabetes mellitus	2.68 (0.84–8.56)	0.094		
NSVT	4.76 (1.81–12.57)	0.002	4.84 (1.55–15.09)	0.007
PMI	0	0		
ICD	4.54 (1.66–12.38)	0.003		
ICD primary prevention	4.50 (1.47–13.77)	0.008		
ICD secondary prevention	2.79 (0.57–13.61)	0.204		
PMI or ICD	3.29 (1.22–8.84)	0.018		
PTSMA	0.13 (0.01–1.03)	0.054		
LAD †	1.06 (0.99–1.12)	0.057		
LVDd †	1.08 (1.01–1.15)	0.022		
LVDs †	1.15 (1.07–1.24)	<0.001	9.39 (2.39–36.93)	0.001
LGE	1.38 (0.54–3.51)	0.499		
CRP ‡	1.69 (1.05–2.73)	0.029		

CI = confidence interval; CRP = C-reactive protein; eGFR = estimated glomerular filtration rate; HR = hazard ratio; ICD = implantable cardioverter defibrillator; LAD = left atrial dimension; LGE = late gadolinium enhancement; LVDd = left ventricular end-diastolic dimension; LVDs = left ventricular diameter at end-systole; LVEF = left ventricular ejection fraction; NSVT = non-sustained ventricular tachycardia; PTSMA = percutaneous transluminal septal myocardial ablation; PMI = pacemaker implantation. †, continuous variable, ‡, cut-off value of 0.82 mg/dL detected by the receiver operating characteristic curve.

**Table 3 jcm-12-05137-t003:** Univariate and multivariable analyses for predictors of LVEF < 60%.

	Univariate Analysis		Multivariable Analysis	
	HR (95% CI)	*p*-Value	HR (95% CI)	*p*-Value
Age	0.98 (0.96–1.00)	0.100		
Male sex	1.99 (1.08–3.68)	0.028		
Atrial fibrillation	3.92 (1.93–7.97)	<0.001	4.11 (1.92–8.81)	<0.001
eGFR †	0.98 (0.96–0.99)	0.007		
Diabetes mellitus	3.78 (1.51–9.46)	0.005		
NSVT	3.45 (1.86–6.41)	<0.001	2.99 (1.54–5.81)	0.001
PMI	0.50 (0.06–4.04)	0.517		
ICD	3.30 (1.57–6.95)	0.010		
ICD primary prevention	2.37 (0.97–5.84)	0.060		
ICD secondary prevention	4.32 (1.39–13.46)	0.011		
PMI or ICD	2.53 (1.25–5.11)	0.010		
PTSMA	1.37 (0.72–2.6)	0.346		
LAD †	1.04 (0.99–1.09)	0.054		
LVDd †	1.04 (0.99–1.09)	0.075		
LVDs †	1.11 (1.05–1.17)	<0.001	3.13 (1.60–6.11)	0.001
LGE	1.65 (0.90–3.02)	0.106		
CRP ‡	2.75 (1.37–5.51)	0.004		

CI = confidence interval; CRP = C-reactive protein; eGFR = estimated glomerular filtration rate; HR = hazard ratio; LAD = left atrial dimension; LGE = late gadolinium enhancement; LVDd = left ventricular end-diastolic dimension; LVDs = left ventricular diameter at end-systole; LVEF = left ventricular ejection fraction; NSVT = nonsustained ventricular tachycardia; PTSMA = percutaneous transluminal septal myocardial ablation; PMI = pacemaker implantation; ICD = implantable cardioverter defibrillator. †, continuous variable, ‡, cut-off value of 0.82 mg/dL detected by the receiver operating characteristic curve.

## Data Availability

This study did not report any data.

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
