# Peer review of "Predictive Factors for Decreasing Left Ventricular Ejection Fraction and Progression to the Dilated Phase of Hypertrophic Cardiomyopathy"

_jcm, 2023, doi:10.3390/jcm12155137_

Round 1
Reviewer 1 Report
This is a significant paper that reports that PTSMA does not deteriorate left ventricular function over the long term.However, I have two comments.
1. In this paper, it is expressed as DHCM, but as described in Document 2, it is a term used only in Japan, so I think it would be better to put the term End-stage HCM somewhere.
2.Although the term "predict" is used, those factors may be the result of impaired left ventricular function, so I think a different term would be less misleading.
I nave no comments.
Author Response
Responses to Comments from Reviewer 1
We are grateful to the reviewer 1 for his/her thoughtful and helpful comments regarding our manuscript. We appreciate your valuable advice to help improve our manuscript. We have made every effort to revise the manuscript in accordance with your comments.
- In this paper, it is expressed as DHCM, but as described in Document 2, it is a term used only in Japan, so I think it would be better to put the term End-stage HCM somewhere.
Response: As per your suggestion, we have added the term End-stage HCM in the revised manuscript (Page 3, Line 45).
Patients with HCM may progress to the dilated phase (DHCM) in other words “End-stage HCM”.
2.Although the term "predict" is used, those factors may be the result of impaired left ventricular function, so I think a different term would be less misleading.
Response: As per your suggestion, we consider the factors we have examined to be both causes and consequences of LVEF reduction. The patients enrolled were selected from those with LVEF ≥50% at baseline, and we do not consider this reflects only the consequences. We have added limitation in the revised manuscript (Page 11, Lines 221-223).
Forth, the predictive factors we examined may be both causes and consequences of LVEF reduction. Thus, the predictive factors mentioned here may actually be more appropriately termed as associated factors.

Reviewer 2 Report
Dear Sir/Madam,
I had the opportunity to act as a reviewer on the recent submission by Ishihara et al. to the Journal of Clinical Medicine.
The authors present an interesting study regarding predictors for progression of hypertrophic cardiomyopathy to dilated cardiomyopathy They conclude that atrial fibrillation, non-sustained ventricular tachycardias, end-systolic left ventricular diameter and CRP are significant predictors.
The manuscript is well structured and written. However, some issues need to be addressed:
1. The main issue is the definition of DHCM – the authors focus on LVEF even if they state that they study the DHCM in the title. There is in manuscript no clear definition of DHCM (i.e., what value). Furthermore, no dichotomization based on left ventricular diameter is made. Furthermore, does the phenotype non-dilated hypokinetic cardiomyopathy (ESC guideline for ventricular arrhythmias) exist in their cohort – how is it represented?
2. A multivariable analysis with n=19 in the group C cannot be made with so many predictors, it is invalid.
3. What patients received a cardiac MRI and an echocardiography at the begin and end of study? Please define clearly.
Best regards,
Author Response
Responses to Comments from Reviewer 2
We are grateful to the reviewer 2 for his/her thoughtful and helpful comments regarding our manuscript. We appreciate your valuable advice to help improve our manuscript. We have made every effort to revise the manuscript in accordance with your comments.
1) The main issue is the definition of DHCM – the authors focus on LVEF even if they state that they study the DHCM in the title. There is in manuscript no clear definition of DHCM (i.e., what value). Furthermore, no dichotomization based on left ventricular diameter is made. Furthermore, does the phenotype non-dilated hypokinetic cardiomyopathy (ESC guideline for ventricular arrhythmias) exist in their cohort – how is it represented?
Response: As per your suggestion, we have added definition of DHCM and left ventricle diamater at post-TTE in the revised manuscript (Pages 4-5, Lines 85-86 and Table 1). The phenotype non-dilated hypokinetic cardiomyopathy was not observed in this study.
DHCM was defined as resulting in decreased LVEF of <50% and LV dilatation.
|
LVDd, mm |
42.83 ± 6.08 |
46.97 ± 5.69 |
52.68 ± 10.59 |
<0.001 |
<0.001 |
|
LVDs, mm |
25.62 ± 4.94 |
32.79 ± 4.63 |
41.37 ± 12.58 |
<0.001 |
<0.001 |
2) A multivariable analysis with n=19 in the group C cannot be made with so many predictors, it is invalid.
Response: As per your suggestion, due to the problem of the number of cases in Groups B and C, we reconsidered that we should limit the multivariate analysis items. In addition, the scoring was limited to three items and the statistics were re-scored, and the results were the same as for the five items, so three items were used. According to the suggestion, we have changed about this in the revised manuscript (Page 2, Lines 33-34, Page 6, Lines 115-117 and Pages 8, Lines 153-157) (Table 2, Table 3, Figure 2 and Figure 3, graphical abstract).
A scoring method based on AF, NSVT, and LVDs, patients with 2 and 3 points had a significantly higher risk of developing DHCM.
Since the number of patients was small in Groups B and C, we limited the selection to top 3 variables for multivariate analysis out of the significant variables in the univariate analysis.
Patients with 2 and 3 points had a significantly higher risk of developing DHCM (odds ratio [OR]: 5.06; 95% confidence interval [CI]: 1.95–13.10; P = 0.001 and OR: 48.21; 95% CI: 8.59–270.76, P < 0.001, respectively) (Figure 2). The risk of declining LVEF (<60%) was significantly associated with 2 and 3 points (OR: 4.00; 95% CI: 2.09–7.69; P < 0.001 and OR: 31.04; 95% CI: 3.65–263.96; P < 0.002, respectively) (Figure 3).
3) What patients received a cardiac MRI and an echocardiography at the begin and end of study? Please define clearly.
Response: Cardiovascular magnetic resonance was imaged at the same time as the initial echocardiogram. As per your suggestion, we have added this section of the materials and methods in the revised manuscript (Page 5, Lines 103-104).
“CMR was imaged at the same time as the initial TTE.”

Reviewer 3 Report
Authors have done a good job.
I have one suggestion:
-If available, please include representative echocardiographic and ECG images, at least one for each study group showing measurements of the key parameters.
-The collected data are upto 2018. Why is it so late in communicating the paper? Can you add some updated patient info? This might increase the impact of the study.
Author Response
Response to Reviewer 3
We are grateful to the reviewer 3 for his/her thoughtful and helpful comments regarding our manuscript. We appreciate your valuable advice to help improve our manuscript. We have made every effort to revise the manuscript in accordance with your comments.
1) If available, please include representative echocardiographic and ECG images, at least one for each study group showing measurements of the key parameters.
Response: As per your suggestion, we attached typical images of echo and ECG in the three groups. (Page 7, Lines 135-136)
Typical images of TTE and ECG in the three groups were showed supplementary Figure.
2) The collected data are upto 2018. Why is it so late in communicating the paper? Can you add some updated patient info? This might increase the impact of the study.
Response: The average observation period for the cases included in this study was 64.9 months. We omitted the most recent cases because we considered it necessary to observe the patients for a longer period.

Round 2
Reviewer 2 Report
Dear Sir/Madam,
Thank you for reviewing the manuscript and addressing the mentioned issues. Indeed, the manuscript was significantly improved, especially after adjusting the multivariable analysis. I have 2 further recommendations:
1. I suggest replacing the word “biomarker” in the figures 2 and 3 (i.e., predictor)
2. The study proposes a novel scoring system for the progression of HCM to its dilated stage. Therefore, I suggest adding a graphical abstract – this helps making the manuscript reader-friendly and gathers more citations on the long term.
Best regards
Author Response
Responses to Comments from Reviewer 2
We are grateful to the reviewer 2 for his/her thoughtful and helpful comments regarding our manuscript. We appreciate your valuable advice to help improve our manuscript. We have made every effort to revise the manuscript in accordance with your comments.
1) I suggest replacing the word “biomarker” in the figures 2 and 3 (i.e., predictor)
Response: As per your suggestion, we have changed from “elevated biomarker” to “corresponding factor” in the revised figures 2 and 3.
2) The study proposes a novel scoring system for the progression of HCM to its dilated stage. Therefore, I suggest adding a graphical abstract – this helps making the manuscript reader-friendly and gathers more citations on the long term.
Response: As per your suggestion, we have added an explanatory document in the revised graphical abstract.
